# CTG-Net: Cross-task guided network for breast ultrasound diagnosis

**Kaiwen Yang**[ID][1,2]*, **Aiga Suzuki**[2], **Jiaxing Ye**[2], **Hirokazu Nosato**[2], **Ayumi Izumori**[3], **Hidenori Sakanashi**[ID][1,2]

**1** Graduate School of Science and Technology, University of Tsukuba, Tsukuba, Japan, **2** National Institute of Advanced Industrial Science and Technology, Tsukuba, Japan, **3** Takamatsu Heiwa Hospital, Kagawa Takamatsu, Japan

* kevin.yang@aist.go.jp

## Abstract

Deep learning techniques have achieved remarkable success in lesion segmentation and classification between benign and malignant tumors in breast ultrasound images. However, existing studies are predominantly focused on devising efficient neural network-based learning structures to tackle specific tasks individually. By contrast, in clinical practice, sonographers perform segmentation and classification as a whole; they investigate the border contours of the tissue while detecting abnormal masses and performing diagnostic analysis. Performing multiple cognitive tasks simultaneously in this manner facilitates exploitation of the commonalities and differences between tasks. Inspired by this unified recognition process, this study proposes a novel learning scheme, called the cross-task guided network (CTG-Net), for efficient ultrasound breast image understanding. CTG-Net integrates the two most significant tasks in computerized breast lesion pattern investigation: lesion segmentation and tumor classification. Further, it enables the learning of efficient feature representations across tasks from ultrasound images and the task-specific discriminative features that can greatly facilitate lesion detection. This is achieved using task-specific attention models to share the prediction results between tasks. Then, following the guidance of task-specific attention soft masks, the joint feature responses are efficiently calibrated through iterative model training. Finally, a simple feature fusion scheme is used to aggregate the attention-guided features for efficient ultrasound pattern analysis. We performed extensive experimental comparisons on multiple ultrasound datasets. Compared to state-of-the-art multi-task learning approaches, the proposed approach can improve the Dice's coefficient, true-positive rate of segmentation, AUC, and sensitivity of classification by 11%, 17%, 2%, and 6%, respectively. The results demonstrate that the proposed cross-task guided feature learning framework can effectively fuse the complementary information of ultrasound image segmentation and classification tasks to achieve accurate tumor localization. Thus, it can aid sonographers to detect and diagnose breast cancer.

**Data Availability Statement:** The data underlying the findings described in this paper are partially available with restriction. THH Dataset cannot be shared publicly because experiments to obtain ultrasound images were conducted with the

participation of subjects on the condition that they are used by National Institute of Advanced Industrial Science and Technology (AIST) and are not disclosed to the public. The UDIAT dataset underlying the results presented in the study are available from (https://helward.mmu.ac.uk/STAFF/m.yap/dataset). The BUSI dataset underlying the results presented in the study are available from (https://scholar.cu.edu.eg/?q=afahmy/pages/dataset).

**Funding:** This study is partly based on results obtained from a project, JPNP20006, commissioned by the New Energy and Industrial Technology Development Organization (NEDO), awarded to HN and HS.

**Competing interests:** The authors have declared that no competing interests exist.

## Introduction

Breast cancer is one of the leading causes of cancer death in women [1]. One important study [2] reported that early diagnosis and treatment of breast cancer can reduce mortality. Breast ultrasound (BUS) imaging is widely used for breast cancer screening because of its safety, low cost, and effectiveness of use on dense breast tissue [3]. However, ultrasound screening for breast cancer is dependent on the judgment of the sonographer, and long hours of repetitive work or inexperience on the part of the sonographer can increase the risk of misdiagnosis. To help sonographers achieve more reliable and accurate diagnoses, the use of a computer-aided diagnosis (CAD) system is particularly important.

A CAD system would be helpful for BUS image analysis to detect lesions induced by breast cancer. As illustrated in Fig 1, a BUS image typically comprises several layers corresponding to the physical structure. The layers are, in order from the top to the bottom of the image, skin and fat, mammary gland, and muscle and rib. Lesions generally occur in the mammary gland layer. The literature [4] has reported three major difficulties in developing CAD systems for BUS images: 1) BUS images are commonly obscure because of speckle noise, low contrast, and artifacts; 2) the breast structure is complex owing to patient differences; and 3) tumors are diverse, leading to wide variability in echo intensity. Thus, the development of CAD systems for the automated interpretation of BUS data has been a long-standing research topic. Various conventional computer vision and machine learning approaches have been used for this task [5]. However, conventional approaches require specialized and skillful extraction of hand-crafted features, making it difficult to handle multiple types of lesions in poor-quality images. In recent years, deep neural networks approaches have continuously demonstrated advanced performance over conventional approaches in various cognitive tasks owing to their powerful capability for automated feature learning. The shift to applying deep learning for BUS image analysis is currently underway [6].

Two common tasks used for BUS image analysis are segmentation and classification, and their outputs differ. The segmentation task involves marking the boundaries of different tissue areas and locating suspicious lesion areas, whereas the classification task involves determining the presence of lesions in the images and the benignity or malignancy of the lesions. Existing studies typically perform segmentation and classification separately. In practice, both tasks are classification in a broad sense (classification is performed at the image level and segmentation at the pixel level). They both make decisions by analyzing patterns in ultrasound images or variations between adjacent tissue regions. If one task is processed effectively, we can expect that

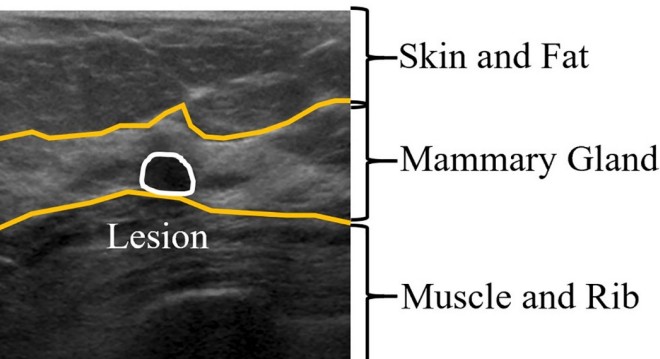

**Fig 1. Schematic of the anatomical structure of a breast ultrasound (BUS) image.**

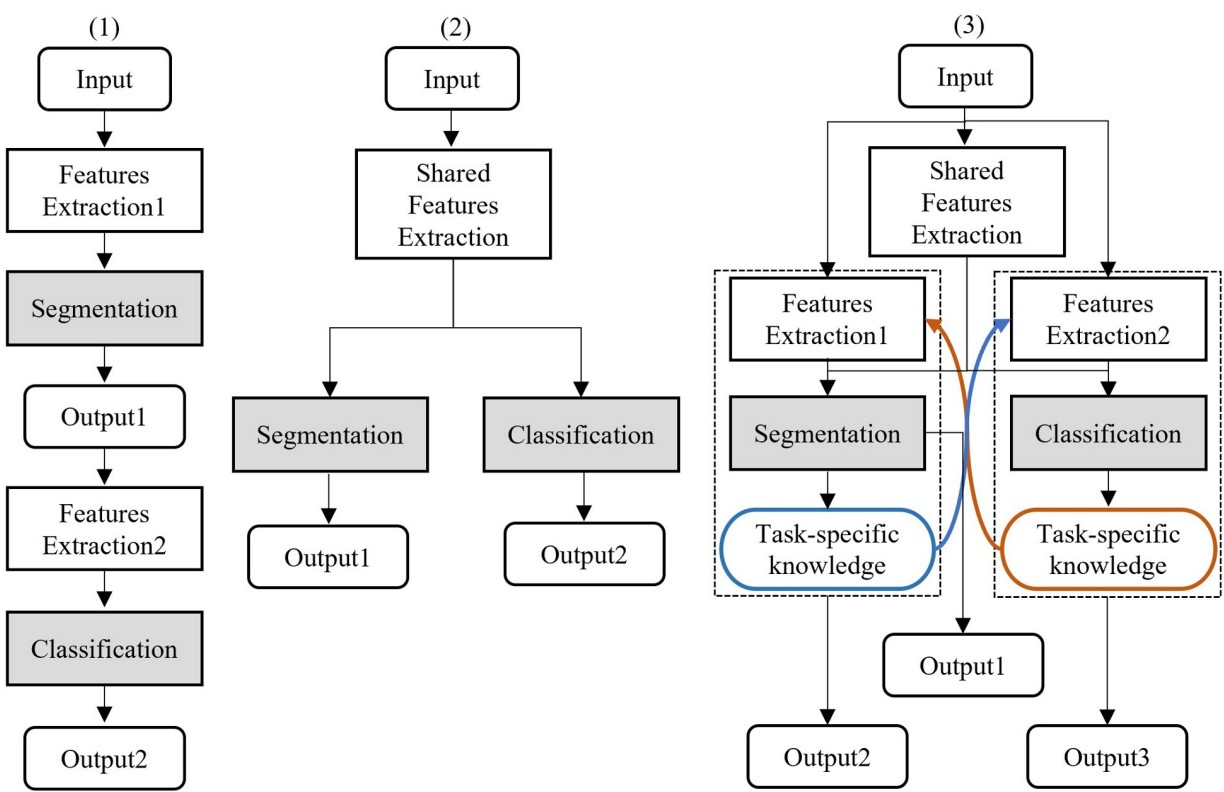

**Fig 2. Illustration of the approaches used for performing BUS image segmentation and classification jointly.** (1) Cascade approach; (2) Multi-task learning approach; (3) Cross-task approach.

useful knowledge could be learned from this task to help the other task. Consequently, performing both tasks together could jointly improve the performance of both.

A straightforward approach to perform segmentation and classification jointly is to first segment candidate lesion regions from the entire BUS image and classify them as either normal or lesioned (benign or malignant); this is called the cascade approach (Fig 2(1)). However, the sequential training of segmentation and classification is necessary in the cascade approach, which leads to additional computational costs. Moreover, the inference in any given stage depends on that in the previous stage. This means that if the previous inference stage fails, the subsequent stage necessarily fails.

The multi-task learning (MTL) approach (Fig 2(2)) can perform segmentation and classification simultaneously without the above-mentioned problems. It is based on the concept of using a shared feature-learning mechanism to extract suitable and efficient feature representations. The shared feature mechanism can learn common knowledge between tasks. However, differences still exist between the participating tasks and we need to consider these differences to suit each actual task. Segmentation and classification can provide complementary information to each other because they are performed under different perspectives. We expect to exploit this information to generate effective task-specific feature representations for further performance improvement.

Consequently, we propose an efficient feature-learning scheme, called the cross-task approach (Fig 2(3)), for considering both the commonality and differences between lesion segmentation and tumor classification tasks. The cross-task approach aims to maximize the

utilization of the knowledge distilled from solving each task to achieve mutual improvement in cognition tasks in BUS images. It has the following two primary advantages. 1) More efficient learning of important features from redundant features can be realized through optimization of the performance with respect to the related tasks. Specifically, the lesion segmentation results can provide localization information for subsequent analysis, i.e., the extracted visual features are anticipated to be applicable to lesion classification. Additionally, the categorization results rendered by classification can further highlight the lesion regions by employing an attention mechanism in the neural network, which can significantly facilitate the segmentation of lesions with finer boundaries. 2) Prior knowledge of breast anatomy obtained from experts can be easily embedded. The knowledge of breast anatomy obtained through segmentation can be used to eliminate over-detection, which could happen in regions where lesions are unlikely to occur, and to improve the sensitivity of judgments in regions where lesions are likely to occur.

**The contributions of this study are summarized as follows**:

- A cross-task approach is proposed to jointly train and mutually improve the correlation tasks in BUS image analysis.

- The cross-task guided network (CTG-Net) is devised to implement cross-task learning for BUS image segmentation and classification.

- Extensive comparative experiment conducted with start-of-the-art classification, segmentation, and multi-task approaches on both private and public datasets are presented. The results obtained on real data validate the proposed approach.

- The proposed approach achieves excellent performance on several private and public datasets with visual differences proving that the proposed approach has good generalization performance and can minimize bias caused by the dataset.

The rest of the paper is organized as follows. Section 2 introduces the related work to the proposed method. Section 3 describes the datasets adopted and explains the overall structure of the proposed method, component units, and loss function. Section 4 presents the experimental setup, evaluation metrics, and experimental results. Section 5 discusses the ablation experiments and failure cases. Finally, this study is concluded in Section 6.

## Related work

Over the last two decades, extensive research has been conducted on CAD diagnostic methods for BUS images to reduce the workload of sonographers and improve the accuracy of diagnosis. The BUS analysis methods can be categorized into two groups: cascade and multi-task learning approaches. Fig 2 presents a conceptual flowchart of the two approaches. In this section, we first review the latest literature and then demonstrate the proposed cross-task guided multi-task approach.

### BUS image analysis using cascade approach

Fig 2(1) gives a conceptual illustration of the cascade approach. In this approach, the suspicious region of the lesion is first segmented, and then classified. Both tasks are performed in a sequential manner. As one typical study of the cascade approach for BUS image analysis, Huang et al. [7] proposed to first segment the tumor region using a conventional level set approach and then performing a benign and malignant classification of the tumor region. In another study, Wang et al. [8] first identified the tumor candidate region using a target

detection network. Subsequently, they performed a benign versus malignant classification of the tumor candidate region.

Studies have also been conducted to tackle individual tasks. For example, various hand-crafted features had been proposed to characterize the shape, texture, position, and orientation of tissue and lesion regions [9, 10]. Statistical classifiers were employed to recognize those patterns (LDA [11], SVM [12], MLP [13]).

Recently, deep convolutional neural networks (DCNNs) have become mainstream methods for BUS image analysis [14]. In this approach, a unified model is adopted to learn feature representations and classify BUS image visual patterns, such as combining DCNNs with image processing techniques to help classification [15]. The latest tweaks to neural networks, such as deep transfer learning [16–19] and the attention model [8, 20, 21], have also been employed. However, the limited availability of data with annotations has been hindering progress. For the BUS image segmentation task, fully convolutional networks [22] and U-Net [23] are the primary backbone models that can favorably characterize local and global information for lesion region segmentation. More recent studies have focused on devising U-Net variants that can further boost the segmentation performance [24–27].

Although the cascaded approach employs a straightforward and reasonable design to unite the segmentation and classification tasks, the sequential process is computationally inefficient. Furthermore, the final classification inevitably fails if the candidate tumor region is incorrectly predicted in the previous stage.

## BUS image analysis using the MTL approach

The MTL approach was introduced to address the issues in the cascade approach. A typical flowchart of this approach is presented in Fig 2(2). The fundamental idea is to exploit the underlying information shared between tasks. For lesion region detection in BUS images, it is evident that the task of mammary gland segmentation and tumor classification have some level of correlation. Thus, it is appropriate to introduce MTL for the application. At its core, MTL aims to learn the generalized representations that are robust and efficient across different tasks. Shared feature learning is achieved by using a particular network design. The front-end layers of the MTL network are designated for feature extraction, followed by parallel independent branches that can exploit specific tasks individually. The feature sharing-based mechanisms can learn multiple tasks simultaneously. With the idea of MTL being widely investigated in natural images [28–30] and other types of medical images (dermoscopy color images [31], abdominal computed tomography scans [32], brain magnetic resonance images [33]), jointly trained BUS image classification and segmentation has also evolved as a major topic. For example, Thome et al. [34] developed an MTL method to perform breast cancer classification and segmentation tasks jointly; the study validated the superiority of the MTL approach. Zhang et al. [35] proposed a soft and hard attention MTL model that integrates BUS images segmentation and classification tasks through soft and hard attention mechanisms, aiming at more efficient use of lesion region information to improve the respective performances. Zhou et al. [36] used the VNet architecture to develop a CAD system that can jointly perform 3D automatic breast ultrasound (ABUS) image classification and segmentation CAD system. They exploited the extracted multi-scale features to improve the image classification task and achieve better results than a single task through an iterative feature refinement strategy. Zhang et al. [37] proposed BI-RADS-Net for explainable BUS CAD based on multi-task learning. The model outputs the probability of class and malignancy of a tumor by performing multiple classification and regression tasks. Cao et al. [38] proposed a multi-task learning method based on label distribution correction for overcoming the problem of insufficient labeled training data.

They performed breast tumor classification task jointly using two labels from different domains of expertise and demonstrated the effectiveness of the method on the collected dataset.

The above survey reveals that the MTL approach could be a promising approach, however, there was little in-depth investigation along this research direction. In contrast to previous studies, our contribution is three-fold, which are as follows. First, it is acknowledged that finding suitable auxiliary tasks plays the most important role for MTL. The tasks should have some level of correlation, otherwise, training on irrelevant tasks can result in negative transfer and deteriorate the performance. To the best of our knowledge, this is the first study to formulate lesion classification and its region segmentation as a multi-task learning problem for BUS image analysis. The two tasks are highly correlated and thus appropriate to be investigated through multi-task learning. Second, to achieve superior performance in lesion classification and segmentation, we adopted the attention mechanism in the proposed neural network design, which enables the network to focus on a few particular aspects that are related to suspicious lesion areas and ignore the rest. In other words, it is an integral building block to generating pixel-wise labels for the lesion region. Third, MTL has been commonly formulated as a minimization of a linear combination of individual tasks' loss functions. The task-specific weights are critical parameters to tune through the learning process. We adopted a self-adjusted scheme to estimate the task-specific weights through optimization, which is more efficient and robust compared to conventional methods such as grid search through cross-validation.

We propose a novel MTL scheme that adopts a particular cross-task guided feature learning design. The core idea is to exploit the intermediate results of individual tasks to recalibrate the joint feature representations and ultimately boost the lesion detection performance. The proposed scheme differs from other MTL scheme in two key features. First, it adopts the attention model to generate task-wise feature representations, which enables more efficient high-level feature learning. Secondly, we aggregate multiple intermediate features for robust lesion region detection. The proposed scheme is illustrated in Fig 2(3) and the details are presented in Section 3.

## Materials and methods

In this section, we first introduce the dataset. Then, we explain the motivation and conceptual design of the cross-task approach. Finally, We present the flowchart of the proposed cross-task guided network (CTG-Net) for BUS image segmentation and classification, subsequentially, introduce each component of the CTG-Net in detail.

### Dataset

In this study, a private dataset was used to evaluate the ability of the deep learning model to distinguish between normal tissue and lesions, and two public datasets were used to evaluate its ability to distinguish between benign and malignant breast cancers. Fig 3 shows sample images from each of the three datasets, and illustrates the average image size in each dataset. We conducted experiments using five-fold cross-validation, which is the most commonly applied validation protocol for empirical analysis, on each of the three datasets. Therefore, we adopted the same criterion to maximize a fair comparison. Notably, lesion images belonging to the same patient were assigned to the same dataset, and the images in each class were divided in equal proportions.

Since this research does not use biological tissues obtained from the human body, it is not necessary to obtain informed consents from the research participants following the "Ethical

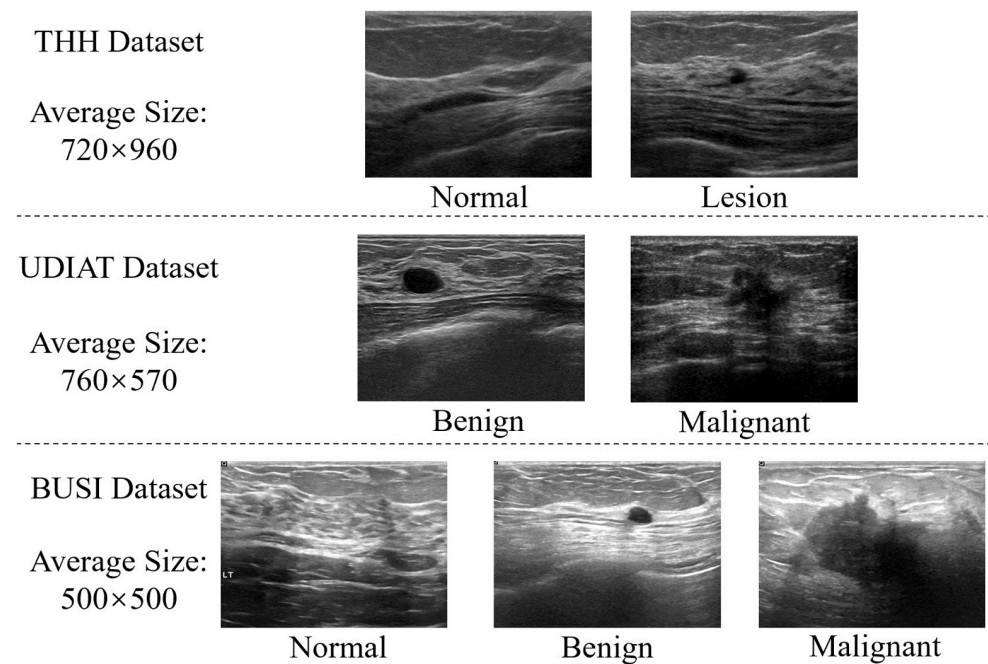

**Fig 3. Sample BUS images from the THH, UDIAT, and BUSI datasets with average image size.**

Guidelines for Medical Research Involving Human Subjects" set by the Japanese Ministry of Health, Labour and Welfare. Therefore, data were collected using the opt-out method rather than obtaining informed consents from the participants. Specifically, the following information regarding the study was posted on the website of Takamatsu Heiwa Hospital, allowing research participants to refuse the following: (1) outline of the study; (2) names of the research institution, head of the research institution, and principal investigator; (3) statement that the research protocol and materials of this study may be obtained or inspected and the method of obtaining or inspecting such materials (the statement also mentions that the information may not be obtained or viewed if it would interfere with the protection of the personal information or intellectual property of research participants); (4) procedures for disclosure of personal information; (5) notification of the purpose of using personal information and the method of handling personal information, including the fact that participants may refuse to have their personal information provided to outside organizations; and (6) contact information for inquiries and complaints. This procedure was approved by the Ethics Committee of Takamatsu Heiwa Hospital on January 18, 2018, and by the Ethics Committee of the National Institute of Advanced Industrial Science and Technology on March 9, 2018 (No. hi2018–0267).

**THH dataset**. The private Takamatsu Heiwa Hospital (THH) dataset was collected from 23 patients examined at Takamatsu Peace Hospital, Japan. From the DICOM multiframe images acquired using a Toshiba Aplio 500 ultrasound system equipped with a Toshiba PLT-1204BX transducer, 2718 normal and 2022 lesion images were collected. The labels between normal and lesion frames and the ground truth of the mammary gland and lesion locations were assigned by an sonographer.

**UDIAT dataset** [39]. The UDIAT dataset comprises 163 images, with 110 benign and 53 malignant. The UDIAT Diagnostic Centre of the Parc Tauli Corporation, Sabadell (Spain) collected the images using the Siemens ACUSON Sequoia C512 system with 17L5 linear array transducer.

**BUSI dataset** [40]. The BUSI dataset contains 133 normal, 487 benign, and 210 malignant images. It was captured by Baheya Hospital using a LOGIQ E9 ultrasound and LOGIQ E9 Agile ultrasound system with ML6–15-D Matrix linear transducer, and the tumor contours were annotated by a specialized sonographer to obtain ground truth.

The THH dataset were obtained by breast surgeons with over 17 years of experience using ultrasound equipment from patients who underwent breast ultrasound examinations at Takamatsu Heiwa Hospital between May 2012 and January 2017 and met the following criteria.

- Selection criteria: (1) Those with findings, such as masses and nonmassive lesions in the mammary glands, or without apparent findings; and (2) those who do not refuse to participate in this study.

- Exclusion criteria: (1) patients who have undergone mastectomy; (2) those substantially thicker or thinner mammary glands or mammary glands with severe mastopathy; (3) those with inadequate samples; and (4) patients who are deemed inappropriate as research participants by the principal investigator.

This procedure was approved by the Ethics Committee of Takamatsu Heiwa Hospital on January 18, 2018, and by the Ethics Committee of the National Institute of Advanced Industrial Science and Technology on March 9, 2018 (No. hi2018–0267).

The UDIAT and BUSI datasets were provided by [39, 40], respectively, and were used under institutional or patient approval. Detailed public datasets access information are provided by the Supporting Information S1 Code.

## Methodology

In this section, we first explain the motivation and conceptual design of the cross-task approach. We then present a flowchart for the proposed CTG-Net for BUS image segmentation and classification. Subsequently, we describe each component of CTG-Net in detail.

BUS image classification and segmentation are two closely related cognitive tasks in breast cancer diagnosis. This research endeavors to investigate both commonalities and differences between the two tasks to achieve complementary information ensemble for better BUS image analysis. Classification and segmentation can achieve mutual complementarity based on two critical evidences: (1) Previous studies [7, 8] demonstrated that segmentation can provide classification with a prior lesion localization information and help classification to exclude interference from regions that hinder judgment. (2) Previous studies [41, 42] performed weakly supervised segmentation using class activation maps for classification and demonstrated that class-specific diagnostic information can highlight lesion regions to help fine segmentation. In this regard, we propose a novel architecture for MTL based on cross-task guided feature learning that can favorably exploit the valuable information conveyed by the "attention feature maps" that are obtained in solving individual tasks. The attention feature map denotes the soft masks generated by the attention module of individual task-specific networks. Moreover, It can be understood to be the probabilistic high-level feature representation of each task, and can be calibrated iteratively through network training. The proposed approach is illustrated in Fig 4. It is noteworthy that the proposed scheme is inherently capable of transferring knowledge between different but related tasks through the link between tasks. Therefore, the features obtained in the shared network, and the soft attention masks, can be learned jointly to maximize the generalization of the features across multiple tasks. Furthermore, a feature fusion layer is incorporated that aggregates the intermediate results of individual tasks for fine-grained lesion detection.

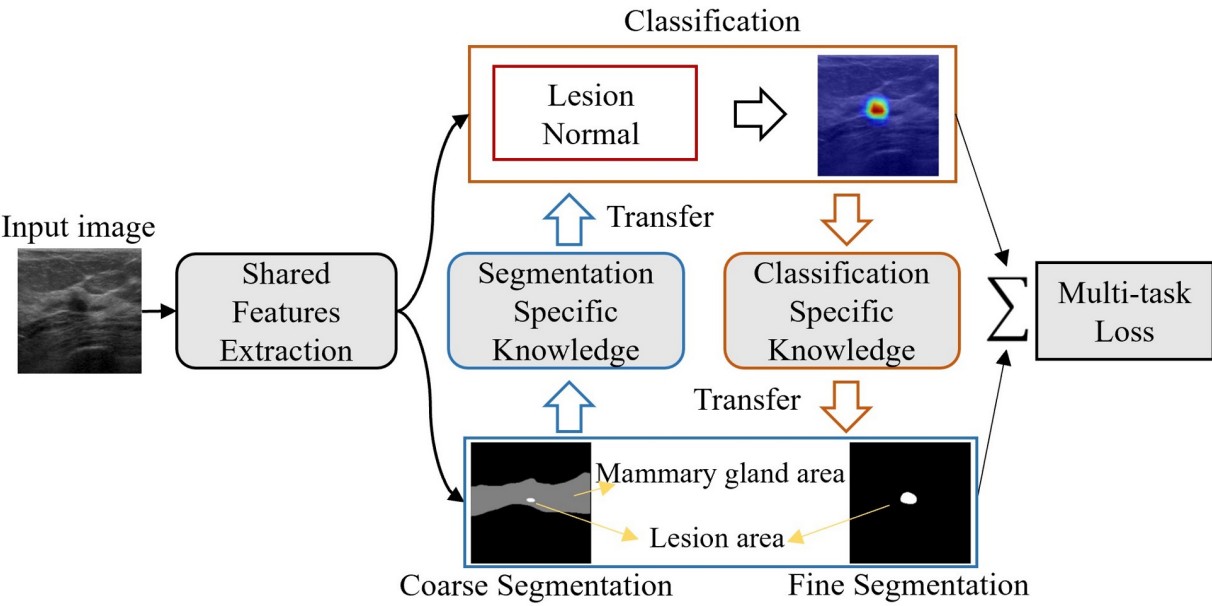

**Fig 4. Illustration of the cross-task approach for achieving mutual improvement between tasks.**

## Network architecture overview

In this section, we elucidate the design of the proposed cross-task guided MTL architecture. A comprehensive flowchart for CTG-Net is presented in Fig 5. The proposed learning architecture comprises four main components: feature extraction unit, coarse segmentation unit, lesion classification unit, and fine segmentation unit. In the ensuing subsections, we describe the functions of each unit in detail.

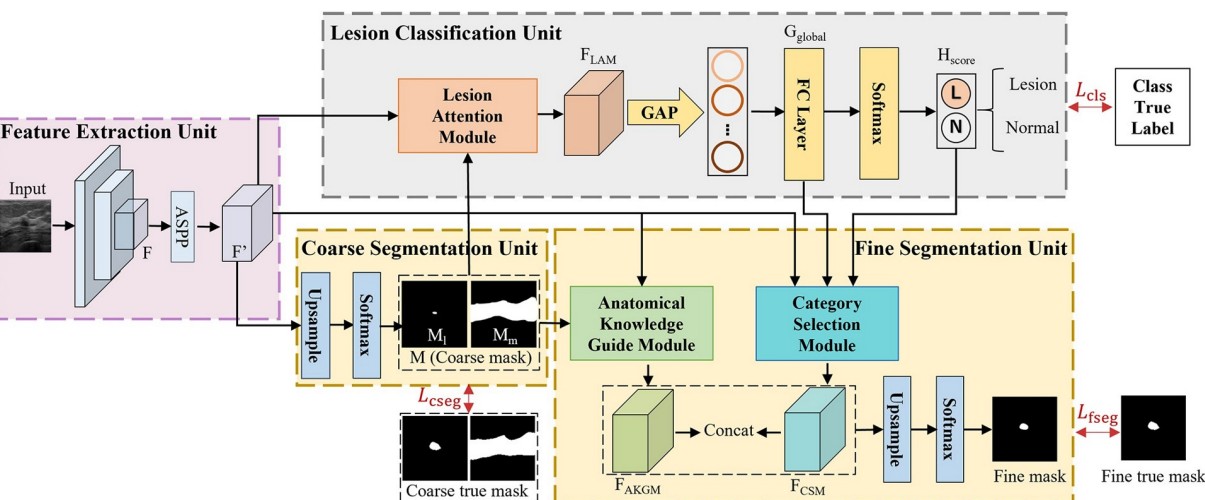

**Fig 5. Architecture of the cross-task guided network (CTG-Net).** The network comprises a shared features backbone and three units for BUS image classification and segmentation. ASPP, Atrous Spatial Pyramid Pooling; GAP, Global Average Pooling; FC, Fully Connected Layer; Concat, Features Channel Concatenation.

## Feature extraction unit

The feature extraction unit utilizes a VGG [43] pre-trained model with ImageNet as the backbone network. Given a single BUS image as input—denoted by $I \in \mathbb{R}^{M \times N}$, where M and N are the height and width of the BUS image, respectively. First, the BUS image is fed into the backbone network that outputs the initial shared feature $F$. Then, we use the Atrous Spatial Pyramid Pooling (ASPP) [44] module to generate shared features $F' = \{F'_i\}_{i=1}^{H \times W}$, $F' \in \mathbb{R}^{D \times HW}$, where $H$ and $W$ are the height and width of the feature maps, and $D$ is the number of channels. The ASPP structure is employed because of its favorable scale-invariant property, which can be helpful in characterizing visual features for small objects.

## Coarse segmentation unit

The coarse segmentation unit uses shared features $F'$ to predict the coarse tissue boundaries $M \in \mathbb{R}^{3 \times HW}$. It consists of three channel binary masks for the mammary gland region $M_m$, coarse lesion location $M_l$, and remaining background $M_b$, respectively. The coarse segmentation unit comprises four upsampling layers, each of which is followed by two convolutional layers. The three channel convolutional layers finally output a coarse mask $M$. Based on the anatomy of the female breast, the mammary gland region is regarded as a potential area of lesion occurrence. The coarse mask $M$ provides critical spatial information about the lesion candidate's presence, which can also facilitate tumor classification. To this end, we further pass the coarse mask $M$, which acts as an intermediate feature, to develop higher-level features that are anticipated to achieve accurate lesion classification.

## Lesion classification unit

The classification unit distinguishes between normal and lesion (benign and malignant) BUS images and obtains task-specific features from the segmentation. Owing to the small proportion of lesioned regions in BUS images and the varying shapes, to obtain reliable attention maps as a guide to locate the lesions, we propose a lesion attention module (LAM), shown in Fig 6. LAM uses the predicted mammary gland region mask $M_m$ and the coarse lesion region mask $M_l$ together as the attention map for the classification task to enhance the feature representation of the lesion. Then, the final feature $F_{\text{LAM}}$ for classification is generated by aggregating the shared features:

$$F_{\text{LAM}} = F' + F_{\text{ma}} + F_{\text{la}}. \tag{1}$$

Here, we perform feature summation to aggregate multiple feature maps. The resultant $F_{\text{LAM}}$ feature can characterize multiple levels of visual information to highlight the lesion region with high accuracy.

Given the predicted mammary gland region mask $M_m$ and the coarse lesion region mask $M_l$, the attention features of the mammary gland region and the lesion region, which are denoted by $F_{\text{ma}}$ and $F_{\text{la}}$ respectively, can be extracted from the following computation:

$$F_{\text{ma}} = F' \odot M_m, \tag{2}$$

$$F_{\text{la}} = F' \odot M_l, \tag{3}$$

where $\odot$ denotes element-wise multiplication. Specifically, we fuse the mammary gland mask and lesion region mask in LAM. The objective is to compensate for the inaccuracy in the initial lesion classification that can incur miss detection of lesion regions with visual ambiguity. The

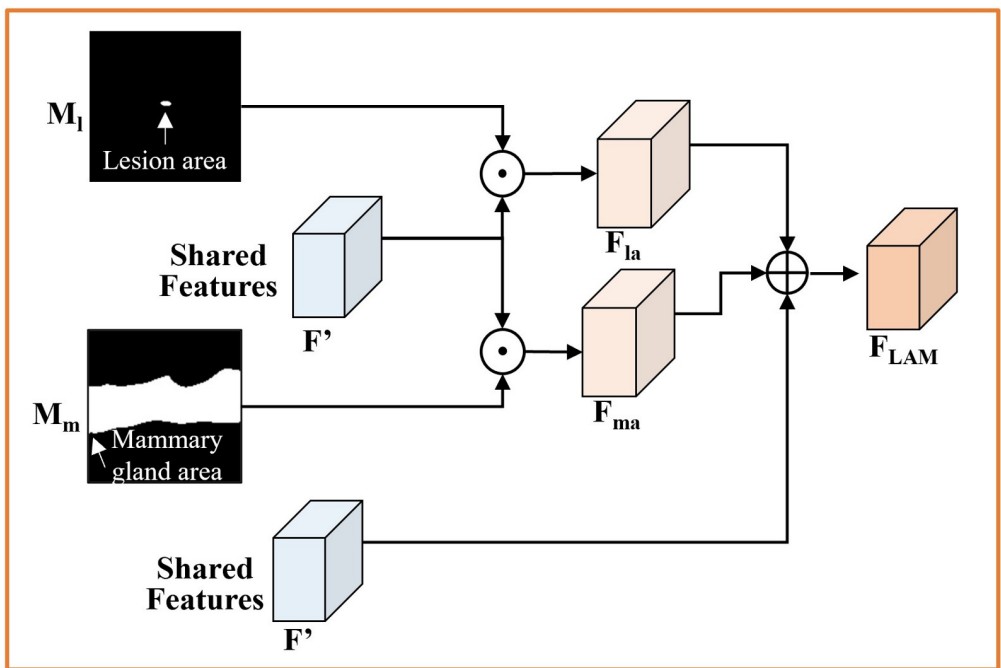

**Fig 6. Schematic of Lesion Attention Module (LAM).** $\odot$ denotes element-wise multiplication and $\oplus$ denotes element-wise addition.

probabilistic representation of attention maps can help remedy the errors in the initial mammary gland region segmentation.

After performing global average pooling (GAP) on $F_{\text{LAM}}$, the produced features are transformed into a global feature vector $G_{\text{global}} = \{g_c\}_{c=1}^{C}$, $G_{\text{global}} \in \mathbb{R}^{D_1 \times C}$ from the classification unit, where $C$ is the number of classes, $g_c$ is the feature vector of the $c$-th category, and $D_1$ is the number of channels. To achieve the classification, it is then fed to the fully connected (FC) layer and the softmax activation function to obtain the prediction scores $H_{\text{score}} = \{h_c\}_{c=1}^{C}$. In addition, the global feature vector $G_{\text{global}}$ and the normalized prediction scores $H_{\text{score}}$ provide information about the lesion category as specific knowledge generated in the classification task. Inspired by the [45], we use the category information to generate attentional features on lesions. They are passed to the fine segmentation unit to guide the segmentation and obtain the final fine segmentation results.

## Fine segmentation unit

The fine segmentation unit segments more accurate lesion regions to obtain task-specific features from the classification. The result of fine segmentation is obtained by concatenating category attention features $F_{\text{CSM}}$ and self-attentive features $F_{\text{AKGM}}$ to produce a better pixel-level predictive feature representation $F_{\text{fine}}$ and then performing an upsampling operation.

$$F_{\text{fine}} = \text{Concat}[F_{\text{CSM}}; F_{\text{AKGM}}], \tag{4}$$

where Concat denotes a concatenation operation on features along the channel dimension. $F_{\text{fine}}$ effectively achieves lesion segmentation by fusing shared features and task-specific features.

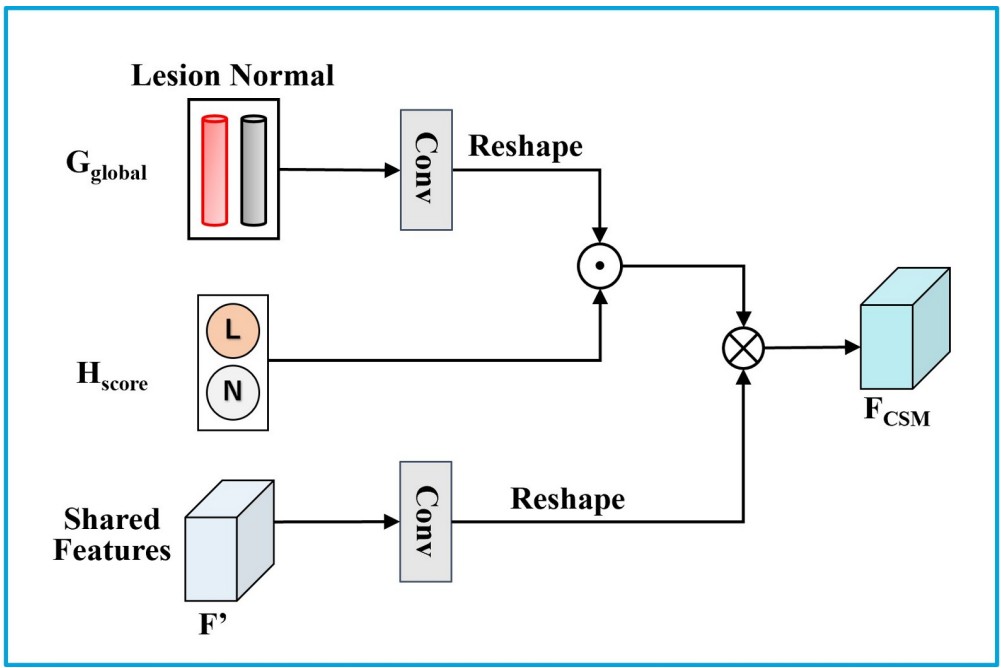

**Fig 7. Schematic of the Category Selection Module (CSM).** ⊙ denotes element-wise multiplication and ⊗ denotes matrix multiplication. Conv indicates that the features have passed through a convolutional layer.

The category attention features $F_{\text{CSM}}$ are obtained by the category selection module (CSM). They are used to improve feature representation by generating category-related attentional features from the classification results task-specific features from the classification. The structure of CSM is shown in Fig 7. The global feature vectors $G_{\text{global}}$ are first transformed with a $1 \times 1$ convolution operation. Subsequently, $G_{\text{global}}$ performs re-weighting of the feature vectors for each class on the category prediction scores $H_{\text{score}}$. Then, the same number of channels as $G_{\text{global}}$ are generated using the $1 \times 1$ convolution operation on the shared features $F'$. Finally, the two perform matrix multiplication to obtain a weighted feature map based on all classes that are the class attention features $F_{\text{CSM}}$, as follows:

$$F_{\text{CSM}} = [\text{Conv}(G_{\text{global}}) \odot H_{\text{score}}{}^{T}]\text{Conv}(F'), \qquad (5)$$

where Conv indicates that the features have passed through a convolutional layer.

The self-attentive features $F_{\text{AKGM}}$, generated by the anatomical knowledge guided module (AKGM), shown in Fig 8, comprise the mammary gland region mask and the self-attentive structure [46] to capture broader contextual information and suppress false positives in regions beyond the mammary gland tissue.

The structure of AKGM is shown in Fig 8. The shared features $F'$ are first fed into the same convolution layer to generate the new feature maps Query ($Q$), Key ($K$), and Value ($V$). $Q$, $K$, and $V$ are extracted from the same features and are used to calculate the correlation between features internally. Then, we use the mammary gland region mask $M_m$ to perform element-wise multiplication with $Q$ and $K$ to obtain $Q'$ and $K'$, respectively. It can obtain more accurate and meaningful spatial location information within the anatomical constraints of the breast. Subsequently, we perform matrix multiplication between the transpose of $K'$ and $Q'$, and apply

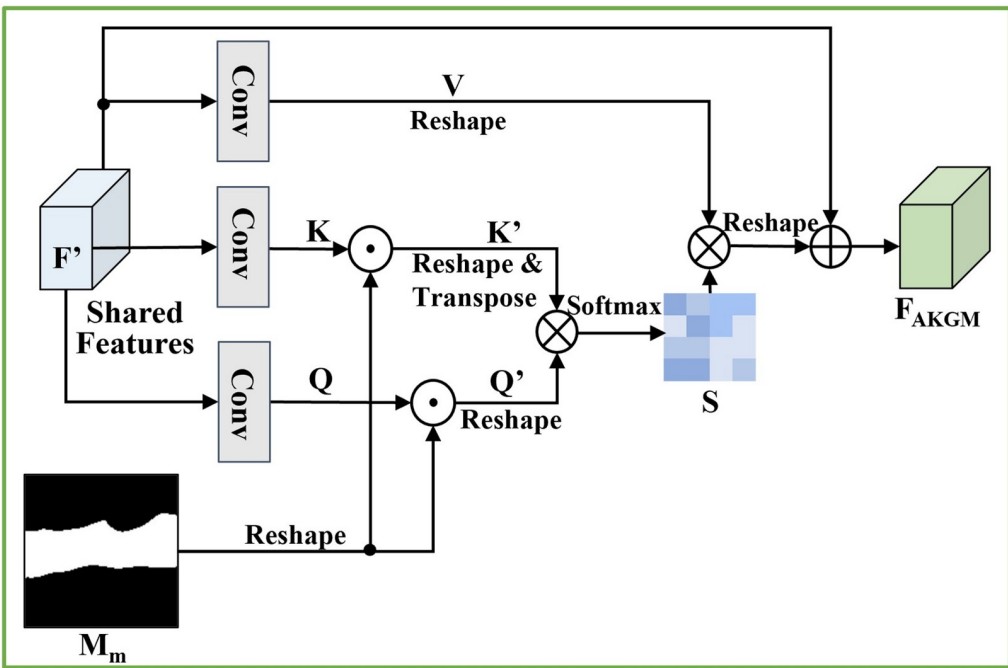

**Fig 8. Schematic of the Anatomical Knowledge Guidance Module (AKGM).** $\odot$ denotes element-wise multiplication, $\otimes$ denotes matrix multiplication and $\oplus$ denotes element-wise addition. Conv indicates that the features have passed through a convolutional layer.

the softmax function to calculate similarity weights $S$:

$$
\begin{aligned}
S &= \text{softmax}((Q' \otimes (K')^T) \\
&= \text{softmax}((Q \odot M_m) \otimes (K \odot M_m)^T).
\end{aligned}
\tag{6}
$$

Then, the similarity weights $S$ and $V$ perform a matrix multiplication. Finally, we multiply it by a scale parameter $\alpha$ and perform an element-wise sum operation with the shared features $F'$ to obtain the self-attentive features with mammary gland constraints $F_{\text{AKGM}}$:

$$
F_{\text{AKGM}} = \alpha(S \otimes V) + F',
\tag{7}
$$

where $\alpha$ is a scalar initialized to zero and is automatically learned during network training. Introducing learnable $\alpha$ makes the self-attention features start learning by relying on the shared features first, and then gradually learning self-attention features containing local to global contextual information as the weights are increased.

## Loss function

For classification, we used cross-entropy loss function training. We employed Dice loss [47], which performs well under the class imbalance problem, as the loss function for the segmentation task since the pixels in the tumor region are severely imbalanced compared to those in the background region. Our CTG-Net is end-to-end, and the overall network is trained by a joint loss function including one cross-entropy loss $L_{\text{cls}}$ in classification unit, two Dice loss $L_{\text{cseg}}$, and $L_{\text{fseg}}$ in the coarse and fine segmentation units. Therefore, the overall loss function is

defined as

$$L_{\text{total}} = \lambda_1 L_{\text{cls}} + \lambda_2 (L_{\text{cseg}} + L_{\text{fseg}}), \tag{8}$$

where $\lambda_1$ and $\lambda_2$ are the task weights for the classification and segmentation, respectively. For the joint optimization of multi-task learning, it is necessary to reasonably assign task weights because of the different training difficulties between tasks. We employed dynamic weight averaging [28] to determine task weights $\lambda_1$ and $\lambda_2$ because this method allows each task to be performed at a similar training rate.

For task $k$, the weights $\lambda_k$ are defined as

$$\lambda_k(t) := \frac{K \exp\left(w_k(t-1)/T\right)}{\sum_i \exp\left(w_i(t-1)/T\right)},$$

$$w_k(t-1) = \frac{L_k(t-1)}{L_k(t-2)}, \tag{9}$$

where $w_k(\cdot)$ calculates the relative descending rate of loss, $i$ is the number of tasks, $w_i(\cdot)$ calculates the relative descending rate of loss for each task, $t$ is an iteration index, and $T$ is a constant for controlling the softness of the task weighting. Further, $T$ is set to two in this study according to experience and experiments, and $L_k(\cdot)$ is the loss value for each iteration.

## Experimental evaluation and results

### Experiment setup

**Training and implementation details.** The proposed framework was implemented in Python based on PyTorch. To accelerate the training process, we used VGG16 [43] pre-trained with ImageNet as the initial state of the feature extraction unit, and other parameters were initialized using random values. We trained the entire network using the Adam [48] optimizer with an initial learning rate of 0.0001 ($\beta_1 = 0.9, \beta_2 = 0.99, \epsilon = 10^{-8}$). Although better optimization methods have been proposed, the Adam optimizer has a simple mechanism and is often used as a standard optimization method by various methods. Therefore, this study uses the Adam optimizer to assess the intrinsic superiority of the proposed approach by checking whether the standard optimization method can also obtain good performance. Furthermore, we trained 100 epochs with a batch size of 16 on the network and observed that the training terminated when the validation set did not improve after 10 consecutive epochs. Algorithm 1 provides the algorithm details to clearly show the optimization process of our proposed method.

**Algorithm 1** Cross-task guided network CTG-Net

```
1: Input: BUS image I with segmentation true mask and class true label
2: Initialization: Feature extraction unit use VGG16 pre-trained with
ImageNet, other parts use random values
3: repeat
4:    F′ ← ASPP(pre-trained VGG16(I)) ▷ ASPP: atrous spatial pyramid
pooling
5:    M(M₁, Mₘ) ← softmax (upsample(F′))
6:    G_global ← FCLayer (GAP(LAM(F′, M))) ▷ GAP: global average pooling
7:    H_score ← softmax(G_global)
8:    F_fine ← Concat(CSM(F′, H_score, G_global), AKGM(F′, Mₘ))
9:            ▷ Concat: concatenation operation
10:   Fine mask ← softmax (upsample(F_fine))
11:   ℓ_cseg, ℓ_cls, ℓ_fseg ← calculate the respective losses using {M,
H_score, Fine mask}
12:   Calculate ℓ_total by Eqs (8) and (9)
```

```
13:    Update model parameter θ by Adam optimizer
14: until validation set not improve after 10 consecutive epochs
15: Output: Parameter set of the CTG-Net model θ
16: function1 LAM(F': shared features, M: coarse mask (M₁, Mₘ))
17:    Fₘₐ ← F' ⊙Mₘ
18:    Fₗₐ ← F' ⊙M₁          ▷ ⊙: element-wise multiplication
19:    Return: F_LAM ← F' + Fₘₐ + Fₗₐ
20: end function1
21: function2 CSM(F', H_score: prediction scores, G_global: global feature
vector)
22:    Return: F_CSM ← [Conv(G_global) ⊙ H_score^T] Conv(F')
23:            ▷ Conv: convolutional layer operations
24: end function2
25: function3 AKGM(F', Mₘ: mammary gland map)
26:    Calculate S using {Mₘ,F'} by Eq (6) ▷ S: similarity weights
27:    V ← Conv(F')
28:    Return: F_AKGM ← α(S⊗V) + F'
29:            ▷ α: learnable parameters, ⊗: matrix multiplication
30: end function3
```

**Evaluation metrics.** *Segmentation task*: To evaluate the image segmentation results, we used four performance metrics: Dice's coefficient (DSC), Jaccard index (JI), true positive rate (TPR), and false positive rate (FPR). Both DSC and JI are measures of similarity between two pixel sets, with the difference being that DSC primarily reflects the average performance for given cases, whereas JI is relatively affected by the worst cases.

$$DSC = \frac{2|A_g \cap A_p|}{|A_g| + |A_p|}, \quad JI = \frac{|A_g \cap A_p|}{|A_g \cup A_p|},$$

$$TPR = \frac{|A_g \cap A_p|}{|A_g|}, \quad FPR = \frac{|(A_g \cup A_p) \setminus A_g|}{|A_g \cup A_p|},$$

where $A_g$ is the pixel set in the lesion region of the ground truth, $A_p$ is the pixel set in the lesion region generated by a segmentation method, and $|\cdot|$ indicates the number of elements of the set. DSC, JI, TPR, and FPR take values in the range [0, 1].

*Classification task*: We evaluated the performance of the classification results using six performance metrics: area under the ROC curve (AUC), accuracy (Acc), sensitivity (Sen), specificity (Spe), precision (Pre), and F1-score (F1).

$$Acc = \frac{TP + TN}{TP + TN + FP + FN}, \quad Sen = \frac{TP}{TP + FN}, \quad Spc = \frac{TN}{TN + FP},$$

$$Pre = \frac{TP}{TP + FP}, \quad F1 = 2 \times \frac{Pre \times Sen}{Pre + Sen},$$

where TP (True Positive) denotes the number of correctly classified lesion images, TN (True Negative) denotes the number of correctly classified normal images, FP (False Positive) denotes the number of normal images incorrectly classified as lesion images, and FN (False Negative) denotes the number of lesion images incorrectly classified as normal images.

## Comparisons with state-of-the-art methods

We compared the segmentation and classification performance of our proposed CTG-Net with several state-of-the-art methods from the perspectives of both single-task and multi-task approaches.

**Table 1. Segmentation and classification performance compared with other state-of-art segmentation models on the THH dataset.**

| (A) | | | | | (B) | | | | | | |
|---|---|---|---|---|---|---|---|---|---|---|---|
| Segmentation | | | | | Classification | | | | | | |
| Method | DSC | JI | TPR | FPR | Method | AUC | Acc | Sen | Spc | Pre | F1 |
| U-Net | 0.49 | 0.36 | 0.37 | **0.03** | AlexNet | 0.81 | 0.82 | 0.73 | 0.90 | 0.81 | 0.77 |
| Attention U-Net | 0.51 | 0.40 | 0.40 | 0.01 | VGG16 | 0.81 | 0.82 | 0.67 | **0.95** | 0.90 | 0.77 |
| Nested U-Net | 0.53 | 0.40 | 0.41 | 0.02 | ResNet18 | 0.82 | 0.85 | 0.70 | **0.95** | 0.90 | 0.79 |
| MA U-Net | 0.66 | 0.54 | 0.56 | 0.04 | DenseNet121 | 0.83 | 0.85 | 0.71 | **0.95** | **0.91** | 0.80 |
| CTG-Net (ours) | **0.78** | **0.67** | **0.78** | 0.13 | CTG-Net (ours) | **0.90** | **0.90** | **0.86** | 0.94 | 0.90 | **0.87** |

The best results are indicated in boldface.

**Comparisons with single-task methods.** *1) Discrimination between normal tissues and lesions.* For the segmentation task, we compared CTG-Net with the widely used BUS image segmentation methods U-Net [23], Attention U-Net [45], Nested U-Net [49], and MAU-Net [25]. The results of the comparison are listed in Table 1(A). The results show that the proposed CTG-Net is poor in terms of FPR, but exhibits significantly better performance in terms of DSC, JI, and TPR. In addition, the qualitative results of CTG-Net and the other methods are presented in columns (c)–(g) of Fig 9. Moreover, CTG-Net still provides more robust performance even for lesions with blurred boundaries and small sizes.

For the classification task, we compared CTG-Net with the commonly used BUS image classification models such as AlexNet [50], VGG16 [43], ResNet18 [51], and DenseNet121 [52]. Based on the results in Table 1(B), CTG-Net significantly outperformed the other methods on all classification metrics. Note that CTG-Net achieved significant improvements for AUC, Acc, Sen, and F1, although ResNet18 and DenseNet121 achieved better performance because of the extraction of deeper semantic features. Columns (h)–(l) of Fig 9 presents visual examples of class activation maps [53]. AlexNet and VGG16 failed to achieve reliable recognition of lesions; instead, they focused more on the background. Conversely, ResNet18 and DenseNet121 roughly localized lesions, but they also focused more on the background. Meanwhile, CTG-Net achieved a more stable and effective focus on the lesion region.

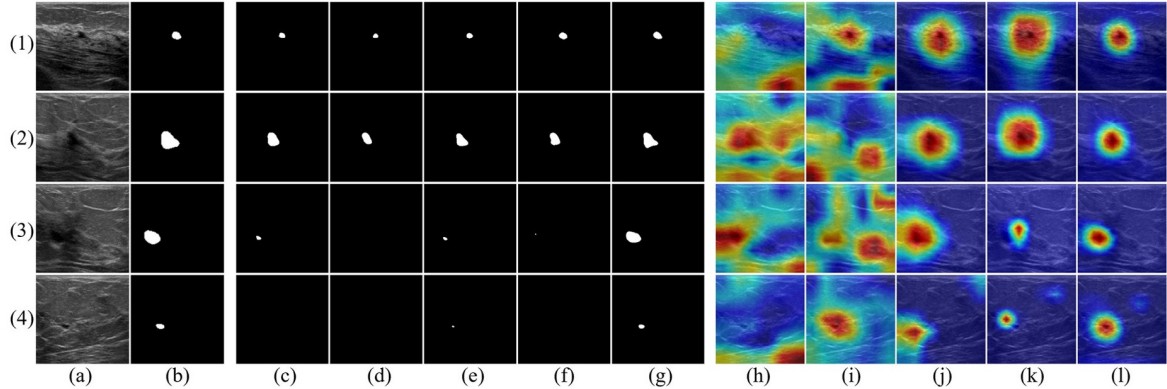

**Fig 9. Visualization of segmentation results and class activation maps for the BUS image private dataset.** (a) BUS images; (b) ground truth (white areas are lesions); (c)–(g) segmentation results are achieved, by U-Net, Attention U-Net, Nested U-Net, MA U-Net, CTG-Net (ours), respectively; (h)–(l) class activation maps are achieved by AlexNet, VGG16, ResNet18, DenseNet121, and CTG-Net (ours), respectively.

**Table 2. Segmentation performance compared with state-of-the-art segmentation models on the UDIAT and BUSI datasets.**

| Dataset | UDIAT | | | | BUSI | | | |
|---|---|---|---|---|---|---|---|---|
| Method | DSC | JI | TPR | tFPR | DSC | JI | TPR | tFPR |
| FCN-AlexNet(2017) | 0.61 | 0.47 | **0.87** | 1.17 | 0.68 | 0.55 | **0.87** | 1.14 |
| SegNet(2017) | 0.71 | 0.60 | <u>0.85</u> | 0.83 | 0.72 | 0.62 | 0.77 | 0.55 |
| U-Net (2015) | 0.75 | 0.65 | 0.78 | 0.41 | 0.73 | 0.63 | 0.77 | 0.56 |
| CE-Net(2019) | 0.72 | 0.61 | 0.74 | 0.48 | 0.73 | 0.64 | 0.77 | 0.64 |
| MultiResUNet(2020) | 0.75 | 0.66 | 0.79 | 0.26 | 0.75 | 0.67 | 0.78 | 0.37 |
| RDAU-NET(2019) | 0.77 | 0.67 | 0.78 | 0.30 | 0.76 | <u>0.68</u> | 0.80 | 0.42 |
| SCAN(2018) | 0.72 | 0.63 | 0.73 | 0.43 | 0.72 | 0.63 | 0.73 | 0.43 |
| DenseU-Net(2019) | 0.72 | 0.64 | 0.74 | 0.43 | 0.72 | 0.64 | 0.74 | 0.43 |
| STAN(2020) | <u>0.78</u> | <u>0.70</u> | 0.80 | 0.27 | 0.75 | 0.66 | 0.76 | 0.42 |
| ESTAN(2020) | **0.82** | **0.74** | 0.84 | <u>0.22</u> | <u>0.78</u> | **0.70** | 0.80 | <u>0.36</u> |
| CTG-Net (ours) | **0.82** | **0.74** | 0.84 | **0.12** | **0.79** | **0.70** | <u>0.82</u> | **0.15** |

The best results are indicated in boldface, and the second-best results are underlined.

*2) Discrimination between benign and malignant.* The proposed method was compared with several state-of-the-art segmentation (see Table 2) and classification (see Table 3) methods on two common BUS image public datasets. The compared methods have different backbone networks and training strategies. Compared with the other state-of-the-art methods on public datasets, we calculated the false-positive rate of CTG-Net using the same method employed in [54] and denoted it the tumor-based false-positive rate (tFPR), which is given by

$$tFPR = \frac{|(A_g \cup A_p) \setminus A_g|}{|A_g|}.$$

The tFPR can be greater than one. tFPR was employed that the tumor area in the BUS image is small and it can better describe the segmentation performance.

For the segmentation task, we compared CTG-Net with the BUS image segmentation methods such as FCN-AlexNet [22], SegNet [55], U-Net [23], CE-Net [56], MultiResUNet [57], RDAU-Net [58], SCAN [59], DenseU-Net [60], STAN [26], and ESTAN [54]. Table 2 summarizes the quantitative results obtained from the two datasets for all segmentation methods.

**Table 3. Classification performance compared with state-of-the-art classification models on the UDIAT and BUSI datasets.**

| Dataset | UDIAT | | | | | | BUSI | | | | | |
|---|---|---|---|---|---|---|---|---|---|---|---|---|
| Method | AUC | Acc | Sen | Spc | Pre | F1 | AUC | Acc | Sen | Spc | Pre | F1 |
| DenseNet121(2016) | 0.82 | 0.83 | 0.52 | **0.98** | **0.93** | 0.67 | <u>0.88</u> | 0.82 | <u>0.77</u> | 0.84 | 0.69 | 0.73 |
| ResNet50(2016) | 0.84 | 0.80 | 0.65 | 0.87 | 0.71 | 0.68 | 0.83 | 0.77 | <u>0.77</u> | 0.76 | 0.60 | 0.67 |
| MLCNN (2019) | - | <u>0.84</u> | <u>0.85</u> | 0.83 | - | - | - | - | - | - | - | - |
| BVANet(2020) | **0.87** | **0.86** | 0.68 | 0.94 | <u>0.86</u> | <u>0.76</u> | **0.89** | <u>0.84</u> | 0.76 | <u>0.88</u> | <u>0.75</u> | <u>0.75</u> |
| BIRADS-SSDL(2020) | 0.70 | 0.79 | 0.50 | <u>0.96</u> | 0.84 | 0.63 | - | - | - | - | - | - |
| CTG-Net (ours) | <u>0.86</u> | **0.86** | **0.87** | 0.85 | 0.75 | **0.80** | **0.89** | **0.90** | **0.84** | **0.93** | **0.85** | **0.85** |

The best results are indicated in boldface and the second-best results are underlined.

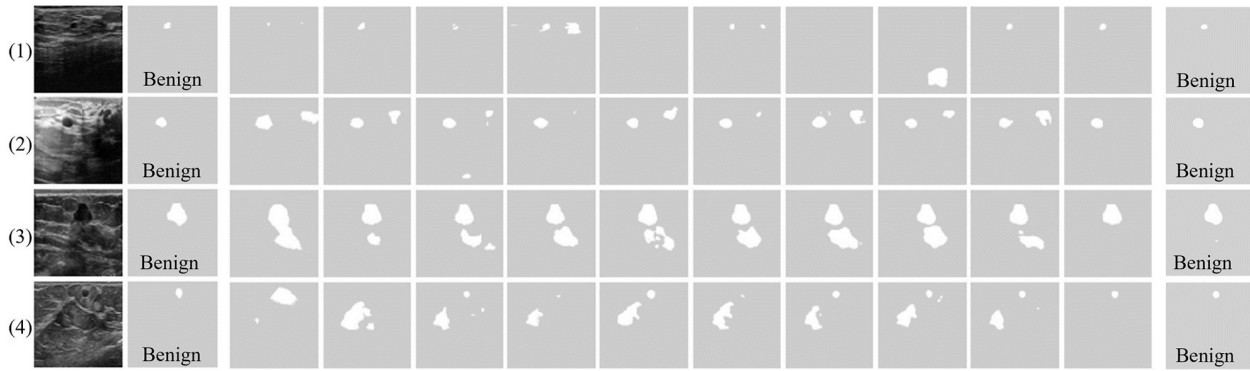

**Fig 10. Examples of results compared with the state-of-the-art methods on BUS image public datasets.** The state-of-the-art results are from [54] and the results for CTG-Net are from this present study. (a) BUS images, (b) ground truth (white areas are lesions and text are true class labels). (c)–(m) Segmentation results achieved by (c) FCN-AlexNet, (d) SegNet, (e) U-Net, (f) CE-Net, (g) MultiResUNet, (h) RDAU-Net, (i) SCAN, (j) DenseU-Net, (k) STAN, (l) ESTAN, and (m) our CTG-Net.

CTG-Net achieved the best performance (highest DSC and JI, and lowest tFPR) on both datasets. Note that CTG-Net achieved lower tFPR than that of the other methods while maintaining a high TPR. Although AlexNet-FCN achieved the best TPR (0.87 and 0.87), it was at the cost of high tFPR (1.17 and 1.14).

We referred to the visualization results of Bryar et al. [54] to depict the qualitative results yielded on the public dataset. Fig 10 presents the results of segmentation compared with our method. Regardless of whether the tumors were small (first and fourth rows) or difficult to distinguish from the background (second and third rows), the CTG-Net results were comparable to those of the state-of-the-art ESTAN model in terms of visualization results. By contrast, the other comparison methods produced high false positives and even failed to detect the tumors. Compared with the overall performance of ESTAN, CTG-Net achieved superior segmentation performance, especially in terms of false-positive suppression.

For the classification task, we compared the classification results with those of the state-of-the-art classification methods, such as DenseNet121 [52], ResNet50 [51], MLCNN [58], BVA-Net [61], and BIRADS-SSDL [62]. Based on the classification results in Table 3, on the UDIAT dataset, the proposed CTG-Net achieved the best values in terms of Acc, Sen, and F1, but did not perform well in terms of Spe and Pre. However, on the BUSI dataset, CTG-Net achieved the best values for all metrics. In terms of the overall performance, the other methods used for comparison sacrificed specificity for higher sensitivity. By contrast, our method maintained high specificity and high sensitivity.

**Comparisons with multi-task learning methods.** We further compared the performance of the proposed model with several state-of-the-art MTL methods [34, 63, 64] for normal tissue and lesion discrimination on the THH dataset to verify the effectiveness of our proposed cross-task guidance. Based on Table 4, CTG-Net has a significant advantage on most of the metrics compared to other methods. These experimental results demonstrate that the proposed method outperforms existing MTL methods in terms of both segmentation and classification performance. This is because the proposed CTG-Net relies on task-specific features extracted from the tasks to achieve a reliable focus on lesions and direct and substantial facilitation effect between tasks.

**Table 4. Segmentation and classification performance compared with state-of-the-art multi-task learning methods on the THH dataset.**

| Method | Segmentation | | | | Classification | | | | | |
|---|---|---|---|---|---|---|---|---|---|---|
| | DSC | JI | TPR | FPR | AUC | Acc | Sen | Spc | Pre | F1 |
| Y-Net (2018) | 0.60 | 0.40 | 0.47 | 0.19 | 0.76 | 0.78 | 0.55 | **0.97** | <u>0.92</u> | 0.69 |
| SC-Net (2019) | <u>0.67</u> | 0.43 | <u>0.61</u> | **0.10** | <u>0.88</u> | <u>0.89</u> | <u>0.80</u> | <u>0.96</u> | **0.93** | <u>0.86</u> |
| Multimix (2021) | 0.63 | <u>0.49</u> | 0.54 | 0.14 | 0.71 | 0.74 | 0.56 | 0.86 | 0.74 | 0.64 |
| CTG-Net (ours) | **0.78** | **0.67** | **0.78** | <u>0.13</u> | **0.90** | **0.90** | **0.86** | 0.94 | 0.90 | **0.87** |

The best results are indicated in boldface, and the second-best results are underlined.

## Discussions

### Ablation study

We also need to determine the importance of each module in the performance improvement on the segmentation and classification tasks. In this section, we discuss the ablation study performed to verify the effectiveness of each module. The ablation study experiments were performed on our collected THH dataset and baseline networks (first row of Table 5) were constructed by removing LAM, AKGM, and CSM from CTG-Net. Table 5 shows that the attentional features generated from the prediction results of the segmentation and classification tasks can mutually facilitate each other, thus improving the performance of both breast lesion segmentation and classification. In the ensuing subsections, we compare the effectiveness of each component in the fine segmentation and classification units in detail.

**Evaluation of CSM and AKGM for segmentation tasks.** The features used for fine segmentation comprise category attentive features from the CSM and self-attentive features with breast anatomical constraints from the AKGM. Table 6 shows the segmentation performance when the two modules are used independently. We can see that AKGM results in few false positives when used alone for fine segmentation. By contrast, the number of false positives increase when CSM alone is used but the segmentation performance is better. This means that

**Table 5. Quantitative results for CTG-Net and baseline (i.e., CTG-Net without any task-specific feature extraction module) on the THH dataset.**

| Method | Segmentation | | | Classification | | |
|---|---|---|---|---|---|---|
| | DSC | TPR | FPR | AUC | Acc | F1 |
| Baseline | 0.68 | 0.68 | 0.16 | 0.84 | 0.86 | 0.81 |
| CTG-Net | **0.78** | **0.78** | **0.13** | **0.90** | **0.90** | **0.87** |

The best results are indicated in boldface.

**Table 6. Ablation study of fine segmentation unit in CTG-Net.**

| AKGM | CSM | DSC | TPR | FPR |
|---|---|---|---|---|
| ✓ | | 0.72 | 0.76 | 0.17 |
| | ✓ | 0.74 | **0.78** | 0.19 |
| ✓ | ✓ | **0.78** | 0.78 | **0.13** |

The best results are indicated in boldface. AKGM denotes Anatomical Knowledge Guidance Module; CSM denotes Category Selection Module.

**Table 7. Ablation study of classification unit in CTG-Net.**

| LAM | | | AUC | Acc | F1 |
|---|---|---|---|---|---|
| $F'$ | $F_{la}$ | $F_{ma}$ | | | |
| ✓ | | | 0.80 | 0.83 | 0.75 |
| ✓ | ✓ | | 0.84 | 0.86 | 0.80 |
| ✓ | | ✓ | 0.83 | 0.84 | 0.79 |
| | ✓ | ✓ | 0.84 | 0.85 | 0.81 |
| ✓ | ✓ | ✓ | **0.90** | **0.90** | **0.87** |

The best results are indicated in boldface. LAM denotes Lesion Attention Module.

segmenting more accurate lesion boundaries increases the risk of false positives. Finally, we obtained the best lesion segmentation performance by concatenating AKGM and CSM, which proves that the mutual combination is effective.

**Evaluation of LAM for classification tasks.** The aim of our proposed LAM is to provide stable and reliable lesion attention to the classification task using the results of the segmentation task, thereby facilitating the classification of normal breast tissue and lesions. We combined three features in LAM: the shared feature $F'$, lesion region attention feature $F_{la}$, and mammary gland region attention feature $F_{ma}$ based on the coarse segmentation results. Table 7 shows the results of the three features under different combination strategies. From the results we can conclude the following: (1) The results of any multiple feature combination strategy are better than those obtained using the shared feature $F'$ alone. (2) The results of the $F' + F_{ma}$ combination strategy are slightly lower than those of the $F' + F_{la}$ combination strategy, which means that the attention using the lesion region can extract more discriminative features to aid classification. (3) The results of the $F_{la} + F_{ma}$ combined strategy were not the best in terms of accuracy owing to the missing shared features resulting in a loss of global information between the two tasks. (4) The fact that the method with all components achieved the best classification performance indicates the efficacy of the proposed feature combination strategy. The results in Table 7 show that the proposed LAM module enables effective focus on important regions in BUS images by using prediction masks, and thus positively boosts the classification.

## Limitations

Despite exhibiting excellent performance for both segmentation and classification, the proposed CTG-Net still makes incorrect predictions in certain specific cases. As shown in Fig 11, false positives and missed detections occurred in images (1) and (2) owing to low contrast, and incomplete segmentation occurred in images (3) and (4) owing to the ambiguity created by blurred boundaries. In addition, further analysis revealed that segmentation labels may not be correctly predicted even if category labels are. For instance, image (1) is correctly predicted as a normal category, but a false positive region is segmented. Image (2) is correctly predicted for the malignant tumor category, but the tumor region is not correctly segment.

The problem of blurred boundaries in few samples remains challenging. Therefore, we intend to introduce the loss of lesion boundaries to address this problem. In addition, for the problem of inconsistent predictions of segmentation and classification tasks, an effective loss function should be designed to supervise the inter-task outputs of each other and ensure the consistency of their predictions.

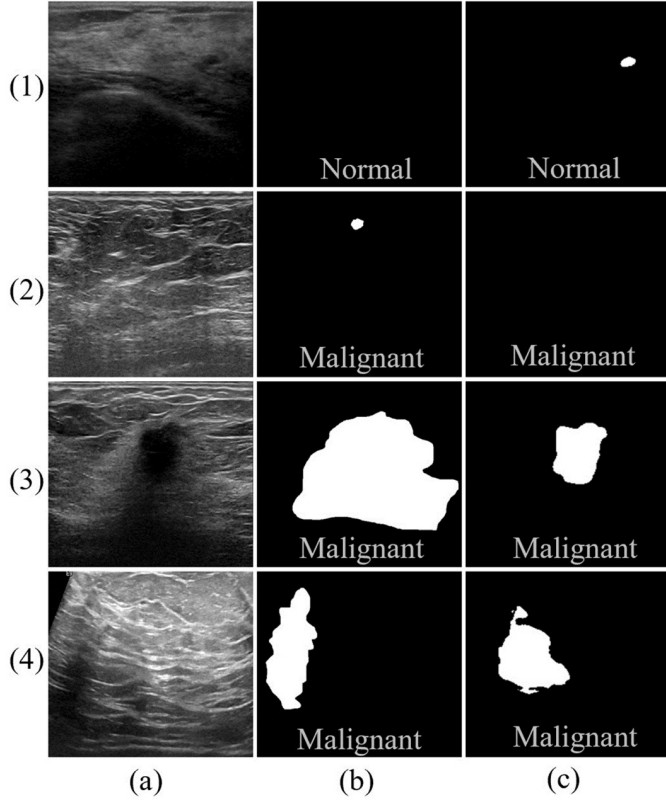

**Fig 11. Illustration of segmentation failure examples.** (a) BUS Images, (b) ground truth, (c) CTG-Net predictions. Image (1) is from the THH dataset, images (2) and (3) are from the UDIAT dataset, and image (4) is from the BUSI dataset. The text represents the category labels.

## Conclusion

In this study, we proposed CTG-Net for implementing the cross-task approach in BUS image segmentation and classification tasks. Unlike conventional MTL approaches, CTG-Net allows the embedding of breast anatomy knowledge to provide prior guidance. Moreover, CTG-Net can generate task-specific features from the prediction results between segmentation and classification tasks to achieve cross-task guided learning for mutual maximum facilitation. The experimental results demonstrate the effectiveness of CTG-Net not only for the recognition of normal and lesion images, but also for the recognition of benign and malignant breast cancers. They also demonstrate that CTG-Net has a significant advantage over existing BUS image segmentation and classification methods. It is anticipated that CTG-Net will improve the accuracy and efficiency of diagnosis by sonographers.

In future work, we will investigate more efficient loss functions to help the proposed model obtain more robust BUS diagnosis results. This includes the introduction of a boundary loss function to help obtain explicit boundaries for different lesions and the investigation of a loss function to help obtain uniform recognition results to ensure consistency of prediction in segmentation and classification.

## Supporting information

**S1 Code. The code of CTG-Net.** This file includes the implementation code of CTG-Net, training and testing details, datasets acquisition information.
(ZIP)

## Acknowledgments

Computational resources of the AI Bridging Cloud Infrastructure (ABCI) provided by the National Institute of Advanced Industrial Science and Technology (AIST) was used.

## Author Contributions

**Conceptualization:** Kaiwen Yang, Hidenori Sakanashi.

**Data curation:** Kaiwen Yang, Ayumi Izumori.

**Formal analysis:** Kaiwen Yang, Aiga Suzuki, Jiaxing Ye.

**Funding acquisition:** Hirokazu Nosato, Hidenori Sakanashi.

**Investigation:** Kaiwen Yang, Aiga Suzuki, Jiaxing Ye, Hirokazu Nosato, Hidenori Sakanashi.

**Methodology:** Kaiwen Yang.

**Project administration:** Hidenori Sakanashi.

**Resources:** Hirokazu Nosato, Ayumi Izumori, Hidenori Sakanashi.

**Software:** Kaiwen Yang.

**Supervision:** Hidenori Sakanashi.

**Validation:** Kaiwen Yang, Aiga Suzuki, Jiaxing Ye, Hidenori Sakanashi.

**Visualization:** Kaiwen Yang.

**Writing – original draft:** Kaiwen Yang.

**Writing – review & editing:** Kaiwen Yang, Aiga Suzuki, Jiaxing Ye, Hirokazu Nosato, Ayumi Izumori, Hidenori Sakanashi.

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
