## [Decision Letter · Decision Letter 0]

5 May 2022

PONE-D-22-12663CTG-Net: Cross-task guided network for breast ultrasound diagnosisPLOS ONE

Dear Dr. Yang,

Thank you for submitting your manuscript to PLOS ONE. After careful consideration, we feel that it has merit but does not fully meet PLOS ONE’s publication criteria as it currently stands. Therefore, we invite you to submit a revised version of the manuscript that addresses the points raised during the review process.

We look forward to receiving your revised manuscript.

Kind regards,

Sathishkumar V E

Academic Editor

PLOS ONE

Journal Requirements:

a) Did participants provide their written or verbal informed consent to participate in this study?

3. Please include information in the Methods section on how the patient data was obtained, and include information on whether the IRB approved this study, and the name of the IRB. Please also include information on how and when participants provided consent for their data to be used in research." 

4. Please note that PLOS ONE has specific guidelines on code sharing for submissions in which author-generated code underpins the findings in the manuscript. In these cases, all author-generated code must be made available without restrictions upon publication of the work. Please review our guidelines at https://journals.plos.org/plosone/s/materials-and-software-sharing#loc-sharing-code and ensure that your code is shared in a way that follows best practice and facilitates reproducibility and reuse.

Reviewers' comments:

Reviewer's Responses to Questions

**Comments to the Author**

1. Is the manuscript technically sound, and do the data support the conclusions?

Reviewer #1: Yes

Reviewer #2: Yes

2. Has the statistical analysis been performed appropriately and rigorously? 

Reviewer #1: Yes

Reviewer #2: Yes

3. Have the authors made all data underlying the findings in their manuscript fully available?

Reviewer #1: Yes

Reviewer #2: Yes

4. Is the manuscript presented in an intelligible fashion and written in standard English?

Reviewer #1: Yes

Reviewer #2: Yes

5. Review Comments to the Author

Reviewer #1: The Research Paper needs the following revisions and is subject for re-review, and after re-review the final decision for the paper will be done:

1. Add in the last lines of Abstract in what %age and in what parameters, the proposed methodology is better as compared to existing techniques and what is the overall analysis of the proposed methodology.

2. Under contributions, add one-two points with regard to experimentation. Add Organization of the paper at end of introduction.

3. Under literature review, it is suggested to add min 15-25 papers which are latest and taken as base for the proposal of methodology, and every paper should be elaborated with what is proposed, what is the novelty and what experimental results are there. At the end of Literature review, highlight in 9-15 lines what overall technical gaps are there in the paper, that led to the design of proposed methodology.

4. Under methods, add the methods, which are used to design the proposed methodology.

5. Add proper system model of the proposed methodology. Add algorithm and flowchart of the proposed methodology.

6. Add Analysis section to the paper.

7. Add some case study based discussion to the paper.

8. Add future scope to the paper.

9. Considering the scope of the paper, add the following references and highlight them properly in the manuscript:

a. Ajantha Devi, V., & Nayyar, A. (2021). Fusion of deep learning and image processing techniques for breast cancer diagnosis. In Deep learning for cancer diagnosis (pp. 1-25). Springer, Singapore.

b. Solanki, A., & Nayyar, A. (2020, December). Transfer Learning to Improve Breast Cancer Detection on Unannotated Screening Mammography. In International Conference on Advanced Informatics for Computing Research (pp. 563-576). Springer, Singapore.

Reviewer #2: 1. Why Adam optimizer is used?

2. References should be updated with recent works related to the proposed study

3. Include the limitations of the study

4. Future work should be added in the conclusion section

5. Provide explanation for wi used in equation 9

6. Why 5 fold cross validation is used?

7. Any hyperparameters associated with the models used in this study?

8. What audience would benefit most from this work?

9. What are the three strongest aspects of this manuscript?

6. PLOS authors have the option to publish the peer review history of their article (what does this mean?). If published, this will include your full peer review and any attached files.

Reviewer #1: No

Reviewer #2: **Yes: **Usha Moorthy

---

## [Author Response · Author response to Decision Letter 0]

19 Jun 2022

Dear reviewer #1, 

Thank you for your valuable advice. After carefully studying your comment, we have finished the revised manuscript and labeled ‘Revised Manuscript with Track Changes’.

Dear reviewer #2, 

Thank you for your thoughtful review comments and questions that help us improve our manuscripts. We have addressed all the issues in the file labeled ‘Revised Manuscript with Track Changes’.

---

## [Decision Letter · Decision Letter 1]

24 Jun 2022

CTG-Net: Cross-task guided network for breast ultrasound diagnosis

PONE-D-22-12663R1

Dear Dr. Yang,

We’re pleased to inform you that your manuscript has been judged scientifically suitable for publication and will be formally accepted for publication once it meets all outstanding technical requirements.

Kind regards,

Sathishkumar V E

Academic Editor

PLOS ONE

Additional Editor Comments (optional):

Reviewers' comments:

Reviewer's Responses to Questions

**Comments to the Author**

1. If the authors have adequately addressed your comments raised in a previous round of review and you feel that this manuscript is now acceptable for publication, you may indicate that here to bypass the “Comments to the Author” section, enter your conflict of interest statement in the “Confidential to Editor” section, and submit your "Accept" recommendation.

Reviewer #1: All comments have been addressed

Reviewer #2: (No Response)

2. Is the manuscript technically sound, and do the data support the conclusions?

Reviewer #1: Yes

Reviewer #2: (No Response)

3. Has the statistical analysis been performed appropriately and rigorously? 

Reviewer #1: Yes

Reviewer #2: (No Response)

4. Have the authors made all data underlying the findings in their manuscript fully available?

Reviewer #1: Yes

Reviewer #2: (No Response)

5. Is the manuscript presented in an intelligible fashion and written in standard English?

Reviewer #1: Yes

Reviewer #2: (No Response)

6. Review Comments to the Author

Reviewer #1: The Revised Paper has incorporated all the comments and now looks perfect in all sense. And now the manuscript stands Accepted with no further revisions.

Reviewer #2: (No Response)

7. PLOS authors have the option to publish the peer review history of their article (what does this mean?). If published, this will include your full peer review and any attached files.

Reviewer #1: **Yes: **Anand Nayyar

Reviewer #2: **Yes: **Usha Moorthy

---

## [Editor Report · Acceptance letter]

3 Aug 2022

PONE-D-22-12663R1 

CTG-Net: Cross-task guided network for breast ultrasound diagnosis 

Dear Dr. Yang:

I'm pleased to inform you that your manuscript has been deemed suitable for publication in PLOS ONE. Congratulations! Your manuscript is now with our production department. 

Kind regards, 

on behalf of

Dr. Sathishkumar V E 

Academic Editor

PLOS ONE